# Alpha/beta power decreases track the fidelity of stimulus-specific information

**Benjamin James Griffiths[1,2], Stephen D Mayhew[1,2], Karen J Mullinger[1,2,3], João Jorge[4], Ian Charest[1,2], Maria Wimber[1,2], Simon Hanslmayr[1,2]***

[1]School of Psychology, University of Birmingham, Birmingham, United Kingdom; [2]Centre for Human Brain Health, University of Birmingham, Birmingham, United Kingdom; [3]Sir Peter Mansfield Imaging Centre, School of Physics and Astronomy, University of Nottingham, Nottingham, United Kingdom; [4]Laboratory for Functional and Metabolic Imaging, École Polytechnique Fédérale de Lausanne, Lausanne, Switzerland

**Abstract** Massed synchronised neuronal firing is detrimental to information processing. When networks of task-irrelevant neurons fire in unison, they mask the signal generated by task-critical neurons. On a macroscopic level, such synchronisation can contribute to alpha/beta (8–30 Hz) oscillations. Reducing the amplitude of these oscillations, therefore, may enhance information processing. Here, we test this hypothesis. Twenty-one participants completed an associative memory task while undergoing simultaneous EEG-fMRI recordings. Using representational similarity analysis, we quantified the amount of stimulus-specific information represented within the BOLD signal on every trial. When correlating this metric with concurrently-recorded alpha/beta power, we found a significant negative correlation which indicated that as post-stimulus alpha/beta power decreased, stimulus-specific information increased. Critically, we found this effect in three unique tasks: visual perception, auditory perception, and visual memory retrieval, indicating that this phenomenon transcends both stimulus modality and cognitive task. These results indicate that alpha/beta power decreases parametrically track the fidelity of both externally-presented and internally-generated stimulus-specific information represented within the cortex.

**\*For correspondence:**
s.hanslmayr@bham.ac.uk

**Competing interests:** The authors declare that no competing interests exist.

## Introduction

Neuronal activity fluctuates rhythmically over time. Often referred to as 'neural oscillations', these rhythmic fluctuations can be observed throughout the brain at frequencies ranging from 0.05 Hz to 500 Hz (*Buzsáki and Draguhn, 2004*). When recording from the human scalp, it is the alpha and beta frequencies (8–12 Hz; 13–30 Hz) that dominate. Alpha/beta activity displays an intimate link to behaviour; engaging in a cognitive task produces a large reduction in the alpha/beta power (amplitude squared). These task-induced power decreases are ubiquitous, and can be observed across species (including humans [*Pfurtscheller et al., 1994*], macaques [*Haegens et al., 2011*], rodents [*Wiest and Nicolelis, 2003*] and cats [*Chatila et al., 1992*]), sensory modalities (including visual [*Pfurtscheller et al., 1994*], auditory [*Krause et al., 1994*], and somatosensory [*Crone et al., 1998*] domains), and cognitive tasks (including perception [*Pfurtscheller et al., 1994*; *Krause et al., 1994*; *Crone et al., 1998*], memory formation/retrieval [*Griffiths et al., 2016*; *Hanslmayr et al., 2009*; *Waldhauser et al., 2016*], and language processing [*Obleser and Weisz, 2012*]). Given their ubiquity, it stands to reason that these decreases reflect a highly general brain process. While numerous domain-general processes have already been ascribed to alpha/beta oscillations (e.g. idling [*Pfurtscheller et al., 1996*]; inhibition [*Jensen and Mazaheri, 2010*; *Klimesch et al., 2007*]), we provide empirical evidence in support of a new perspective: alpha/beta power decreases are a proxy for information processing.

To successfully process information about a stimulus, the brain must be capable of elevating the signal of said stimulus above the noise generated by ongoing neuronal activity (*Harris and Thiele, 2011*). In situations where the ongoing spiking of a large population of neurons is correlated, this is problematic (*Averbeck et al., 2006*). Mass synchronised spiking generates noise that conceals the comparatively small neuronal signal evoked by the stimulus (see *Figure 1a*), rendering momentary changes in sensory input undetectable (*Busch et al., 2009*) and responses to temporally-extended changes unreliable (*Goard and Dan, 2009*). Reducing these neuronal 'noise correlations', therefore, can boost the signal-to-noise ratio of an evoked neuronal response to a stimulus. Indeed, numerous studies have demonstrated that the decorrelation of task-irrelevant neuronal firing accompanies engagement in cognitive tasks (*Goard and Dan, 2009*; *Poulet and Petersen, 2008*; *Mitchell et al., 2009*; *Churchland et al., 2010*). Given that these noise correlations show a strong positive correlation with the local field potential (LFP) (*Cui et al., 2016*), one may speculate that task-induced reductions in alpha/beta LFP (*Haegens et al., 2011* ) are (to some degree) a marker of the reduction of noise correlations. Such a hypothesis would explain why reductions in alpha/beta power are associated with the successful execution of a wide range of cognitive tasks, from visual perception (*Pfurtscheller et al., 1994*) to memory retrieval (*Michelmann et al., 2016*).

Here, we test the hypothesis that alpha/beta power decreases are a proxy for information processing (*Hanslmayr et al., 2012*). Specifically, we predict that as the amount of stimulus-specific information within the cortex increases, concurrently-recorded measures of alpha/beta power will decrease. Twenty-one participants took part in an associative memory task whilst simultaneous EEG-fMRI recordings were obtained (see *Figure 1b*). On each trial, participants were presented with one of four videos (and on alternating blocks, one of four melodies), followed by a noun, and asked to pair the two. Later, participants were presented with the noun and asked to recall the associated video/melody (which would lead to the reinstatement of stimulus-specific information about the video/melody; *Staresina et al., 2012*). We first conducted representational similarity analysis (RSA) on the acquired fMRI data to quantify the relative distance between neural patterns of matching and differing videos/melodies. This provides a data-driven and objective measure of stimulus-specific information present during a single trial. We then derived alpha/beta power from the concurrently recorded EEG and correlated the observed power with our measure of stimulus-specific information on a trial-by-trial basis. Foreshadowing the results reported below, we found that post-stimulus alpha/beta power decreases negatively correlated with the amount of stimulus-specific information observed in the cortex. Importantly, we find evidence for this during both the perception and retrieval of these videos, as well as during the perception of the melodies, providing conceptual replication of our results and supporting the modality- and task-general nature of our hypothesis.

## Results

### Detecting stimulus-specific information in BOLD patterns

Our first step was to derive a measure of stimulus-specific information (that is, information unique to each of the four repeatedly-presented videos/melodies) from the acquired fMRI data (for univariate analyses, see *Figure 2—figure supplement 1*). To this end, we used searchlight-based representational similarity analysis (RSA) to quantify the overlap in BOLD patterns for matching videos/melodies, and contrasted this against the overlap between with the three other repeated videos/melodies (analysis was always restricted to within a modality, at no point were visual patterns contrasted against auditory patterns). We interpret the difference in overlap between matching and differing videos/melodies as the amount of stimulus-specific information present on a single trial, under the assumption that any similarity that can only be explained by matching stimuli represents information specific to that stimulus. To evaluate whether the quantity of stimulus-specific information was meaningful within a searchlight, the observed measure of information was contrasted against the null hypothesis (i.e. that BOLD pattern overlap for matching videos is the same as BOLD pattern overlap for differing videos) in a one-sample, group-level t-test.

For the perceptual tasks, stimulus-specific information was quantified by computing the representational distance between every pair of perceptual trials. During visual perception, whole-brain searchlight analysis revealed a significant increase in stimulus-specific information relative to chance bilaterally in the occipital lobe ($p_{FWE}$ <0.001, k = 9911, peak MNI: [x = −30, y = −57, z = −2],

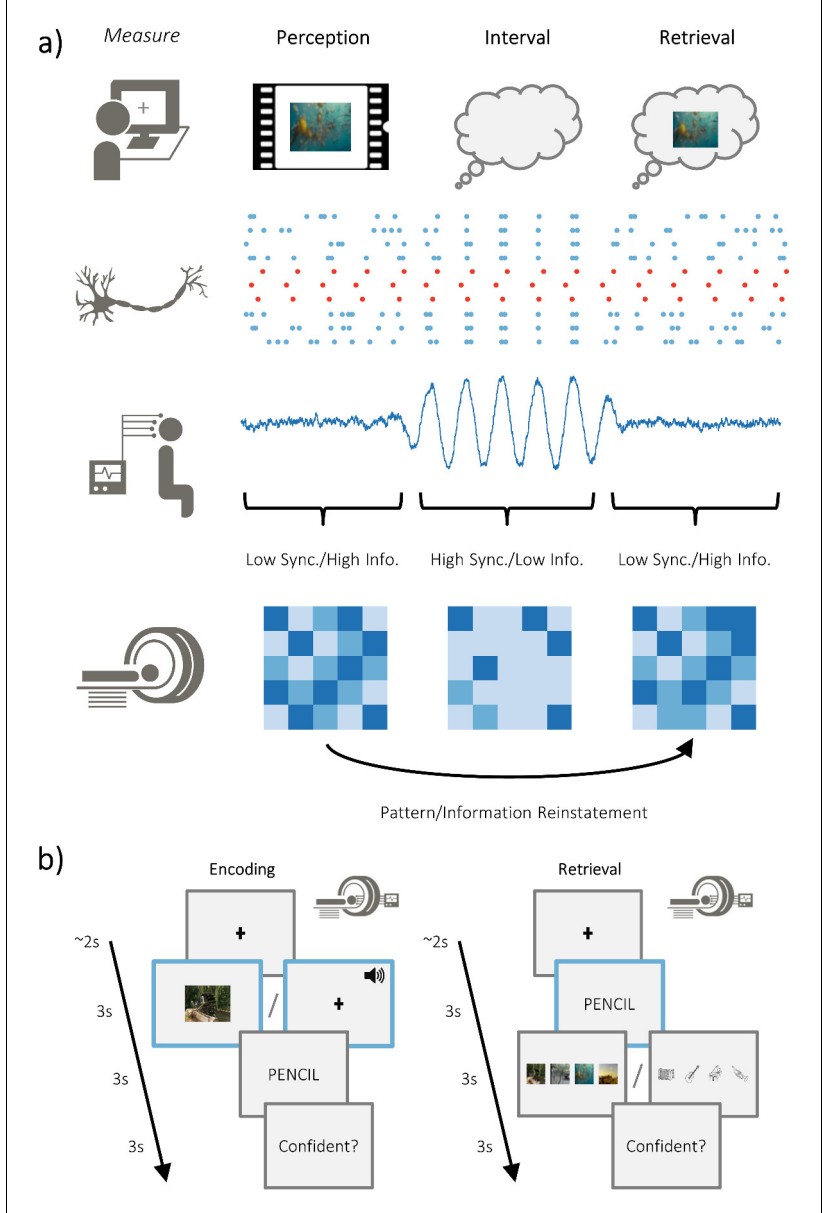

**Figure 1.** Overview of hypotheses and paradigm. (a) The brain is capable of representing stimulus-specific information through neural patterns that are consistent regardless of whether the stimulus is externally or internally generated (i.e. perceived or retrieved; top). On a neuronal level, populations that code for the stimulus (in red) need to generate signal greater than ongoing neuronal noise (in blue). When the neuronal noise correlates (i.e. arises at the same time; during the 'interval'), the signal-to-noise ratio is reduced and stimulus specific information is limited. These noise correlations may be reflected in macroscopic measures of electrophysiological activity, where periods of highly synchronised firing is accompanied by periods of high amplitude activity. Under this assumption, high amplitude activity would reflect an attenuation of the processing of stimulus-specific information. Stimulus-specific information can be measured using fMRI to look at pattern similarity during perception and pattern reinstatement during memory retrieval. (b) Participants completed an associative memory task while undergoing simultaneous EEG-fMRI recordings. Participants were asked to vividly associate a video/melody with a word, and then rate how plausible (i.e. believable) the imagined association was. Later, they were cued with the word and tasked with recalling the associated video/melody. After selecting the associated video/melody, they were asked to judge how confident they felt about their decision. The modality of the dynamic stimuli alternated at the end of each block (counterbalanced across participants).

Cohen's $d_z$ = 1.79) [see *Figure 2a*]. This corroborates the findings of numerous early studies (for example *Chen et al., 2017*; *Baldassano et al., 2017*) which indicate that the occipital lobe is critical in the representation of dynamically-unfolding visual information. A frontal central cluster ($p_{FWE}$ <0.001, k = 113, peak MNI: [x = 12, y = −16, z = 50], Cohen's $d_z$ = 0.28) and a left temporal cluster ($p_{FWE}$ = 0.003, k = 64, peak MNI: [x = −48, y = −1, z = 18], Cohen's $d_z$ = 0.81) were also uncovered. The former may reflect goal-directed tracking of the visual stimulus (*Holroyd et al., 2018*), while the latter may reflect high-level semantic representations of the stimulus (*Visser et al., 2010*; *Rice et al., 2015*). During auditory perception, whole-brain searchlight analysis revealed a significant increase in stimulus-specific information relative to chance bilaterally in the temporal lobe (left temporal: $p_{FWE}$ <0.001, k = 698, peak MNI: [x = −57, y = −37, z = 10], Cohen's $d_z$ = 0.86; right temporal: $p_{FWE}$ <0.001, k = 859, peak MNI: [x = 60, y = −25, z = 10], Cohen's $d_z$ = 1.20) [see

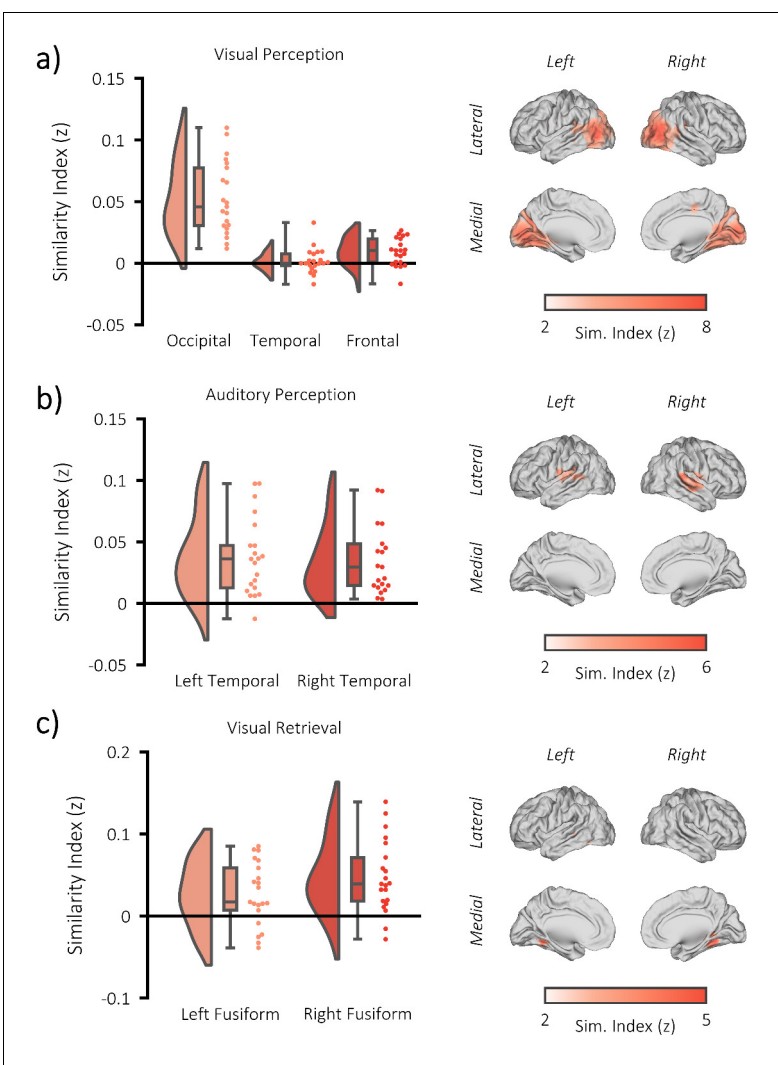

**Figure 2.** fMRI RSA searchlight analysis. (a) raincloud plot (left) depicting the degree to which matching and differing stimuli could be distinguished from one another during visual perception, per participant (single dots), within the significant cluster, and brain map (right) depicting the cluster where matching and differing stimuli could be distinguished from one another. (b) raincloud plot (left) and brain map (right) for stimulus discriminability during auditory perception. (c) raincloud plot (left) and brain map (right) for stimulus discriminability during visual memory retrieval.

The online version of this article includes the following figure supplement(s) for figure 2:

**Figure supplement 1.** Univariate BOLD contrasts.

**Figure supplement 2.** Additional fMRI RSA searchlight analysis.

*Figure 2b*]. These results demonstrate that stimulus-specific information is represented within the cortex during both visual and auditory perception, and identify the regions where a meaningful measure of visual/auditory stimulus-specific information can be derived for our central analysis.

For the retrieval task, stimulus-specific information was quantified by comparing every retrieval pattern with every perceptual pattern. This approach is sensitive to the reinstatement of veridical information about a successfully recalled stimulus (*Staresina et al., 2012*). As we would not anticipate that any stimulus-specific information is present in the BOLD signal when the correct stimulus is not recalled, this analysis was restricted to trials where the paired associate was successfully recalled (see *Figure 2—figure supplement 2* for analysis of the forgotten pairs). During visual memory retrieval, whole-brain searchlight analysis revealed a significant increase in reinstated stimulus-specific information relative to chance in the right fusiform gyrus ($p_{FWE}$ <0.001, k = 313, peak MNI: [x = 30, y = −46, z = −14], Cohen's $d_z$ = 1.07) and left fusiform gyrus ($p_{FWE}$ <0.001, k = 456, peak MNI: [x = −45, y = −37, z = −6], Cohen's $d_z$ = 0.69) [see *Figure 2c*]. Notably, this effect was not driven by the presentation of video stills that followed the presentation of the retrieval cue (see *Figure 2—figure supplement 2*). These results demonstrate that stimulus-specific information is reinstated during the retrieval of visual information, and provide a region of interest that yields a meaningful measure of stimulus-specific information for the central analysis of the memory task.

No significant cluster was identified during auditory memory retrieval (left frontal cluster: $p_{FWE}$ = 0.153, k = 28, peak MNI: [x = −39, y = 44, z = 14], Cohen's $d_z$ = 0.36; see *Figure 2—figure supplement 2*). It is unclear why we could not find evidence for retrieved auditory information, but it may be explained by the fact that memory performance was substantially worse for the auditory stimuli (52.9%) when compared to visual stimuli (73.8%; t(20) = 7.13, p<0.001). Poor memory performance meant that fewer trials could be included in the similarity analysis, limiting the statistical power of the analysis. With no measure of stimulus-specific information for retrieved auditory memories, we could not test our central hypothesis on this portion on the data.

## Alpha/beta power decreases accompany task engagement

We then measured the degree to which alpha/beta power drops during task engagement. As such an effect is perhaps the most ubiquitous effect in studies of task-related scalp EEG activity, it provides a strong benchmark for the quality of our EEG data (which has the potential for distortion by MRI-related artifacts; *Fellner et al., 2016*). For both the perceptual and retrieval trials, the time-series of every source-reconstructed virtual EEG electrode was decomposed into alpha/beta power using 6-cycle Morlet wavelets and baseline-corrected using z-transformation. Alpha/beta power was defined as power between 8 and 30 Hz, as two previous experiments (*Michelmann et al., 2016*; *Griffiths et al., 2019*) using this paradigm found that this frequency range best described task-related decreases in the alpha and beta frequencies. In the first instance, post-stimulus power (500 to 1500 ms) was contrasted against pre-stimulus power (−1000 to −375 ms) in a cluster-based, permutation t-test. We found a significant decrease in alpha/beta power following visual stimulus presentation (p<0.001, Cohen's $d_z$ = 0.95; see *Figure 3a–d*) and auditory stimulus presentation (p=0.012, Cohen's $d_z$ = 0.53) tasks. Power decreases evoked by the visual stimuli were predominately observed in the occipital lobe, while power decreases evoked by the auditory stimuli were observed in the parietal and temporal lobes (see *Figure 3d*).

We then asked whether alpha/beta power decreases are not only predictive of task engagement, but also task success. In other words, is the reduction in post-stimulus alpha/beta power greater when memories are successfully recalled? As in the above paragraph, this is not a novel idea and has been demonstrated many times prior (*Michelmann et al., 2016*). Nevertheless, we wanted to further demonstrate the robustness of our acquired EEG data. To this end, the post-stimulus alpha/beta power (500–1500 ms; 8–30 Hz; matching previously-reported windows of retrieval-related memory effect; *Michelmann et al., 2016*) for remembered trials was contrasted with that of forgotten trials in a cluster-based, permutation t-test. Matching earlier reports, we found a significant reduction in alpha/beta power for recalled pairs, relative to forgotten pairs (p=0.017, Cohen's $d_z$ = 0.56; see *Figure 3* and *Figure 3—figure supplement 1*). These power decreases were localised to the late visual ventral stream (including the region within the fusiform gyrus where stimulus-specific information could be identified), as well as other parts of the memory network (*Rugg and Vilberg, 2013*) (including the medial temporal lobe and medial prefrontal cortex; see *Figure 3d*). No region exhibited an increase in alpha/beta power during this window.

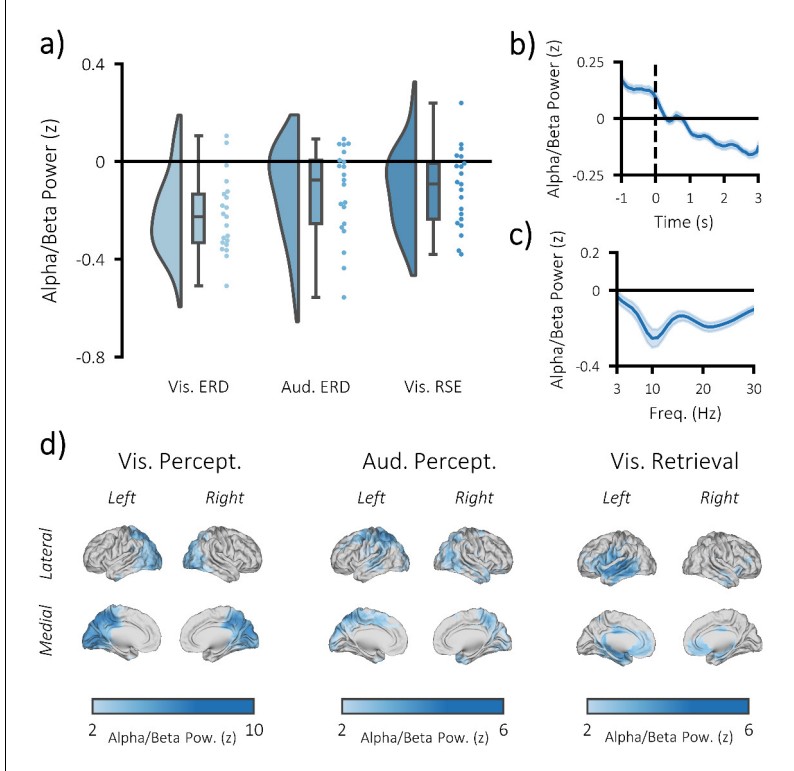

**Figure 3.** Task-induced decreases in post-stimulus alpha/beta power. (a) Raincloud plot displaying event-related decreases in power during visual perception (left) and auditory perception (middle), and memory-related decreases in power during visual retrieval (each dot represents a single subject). (b) time-series of alpha/beta (8–30 Hz) power change over time across all three tasks (see *Figure 3—figure supplement 1* for each task individually). The dark line indicates the mean across participants; the shaded error bar represents standard error of the mean [N.B. as the electrodes were chosen because they belonged to the significant cluster, these figures are for descriptive purposes only and should not be used for statistical inference]. (c) frequency spectrum of post-stimulus power across all three tasks (500–1500 ms; referenced to −1000 to −375 ms pre-stimulus power). The dark line indicates the mean across participants; the shaded error bar represents standard error of the mean (see *Figure 3—figure supplement 1* for each task individually). (d) brain maps of the event-related (left/middle) and memory-related (right) differences in alpha/beta power.

The online version of this article includes the following figure supplement(s) for figure 3:

**Figure supplement 1.** Time-series (top) and difference in spectral power (bottom) for visual perception (left; post-stimulus >pre stimulus), auditory perception (middle; post-stimulus >pre stimulus) and visual memory retrieval (right; hits > misses).

## Alpha/beta power decreases track the fidelity of stimulus-specific information

We then addressed our central question: do alpha/beta power decreases parametrically track the fidelity of stimulus-specific information? For each participant, a single trial measure of stimulus-specific information was computed by comparing the trial pattern within the region of interest (i.e. the significant clusters identified in the fMRI searchlight analysis; see *Figure 2*) to patterns of matching and differing videos/melodies. For the perceptual data, this approach involved computing the representational distance for every pair of perceptual trials. These distances were then correlated with a unique model for each trial that stated representational distance for stimuli matching the stimulus presented would be zero and representational distance for stimuli differing from the stimulus presented on this trial would be one. The resulting correlation coefficient was Fisher z-transformed to provide a normally-distributed metric of stimulus-specific information for each trial. Alpha/beta power within the region that housed stimulus-specific information was calculated and averaged over virtual electrodes, frequency and time. The metric of stimulus-specific information was then

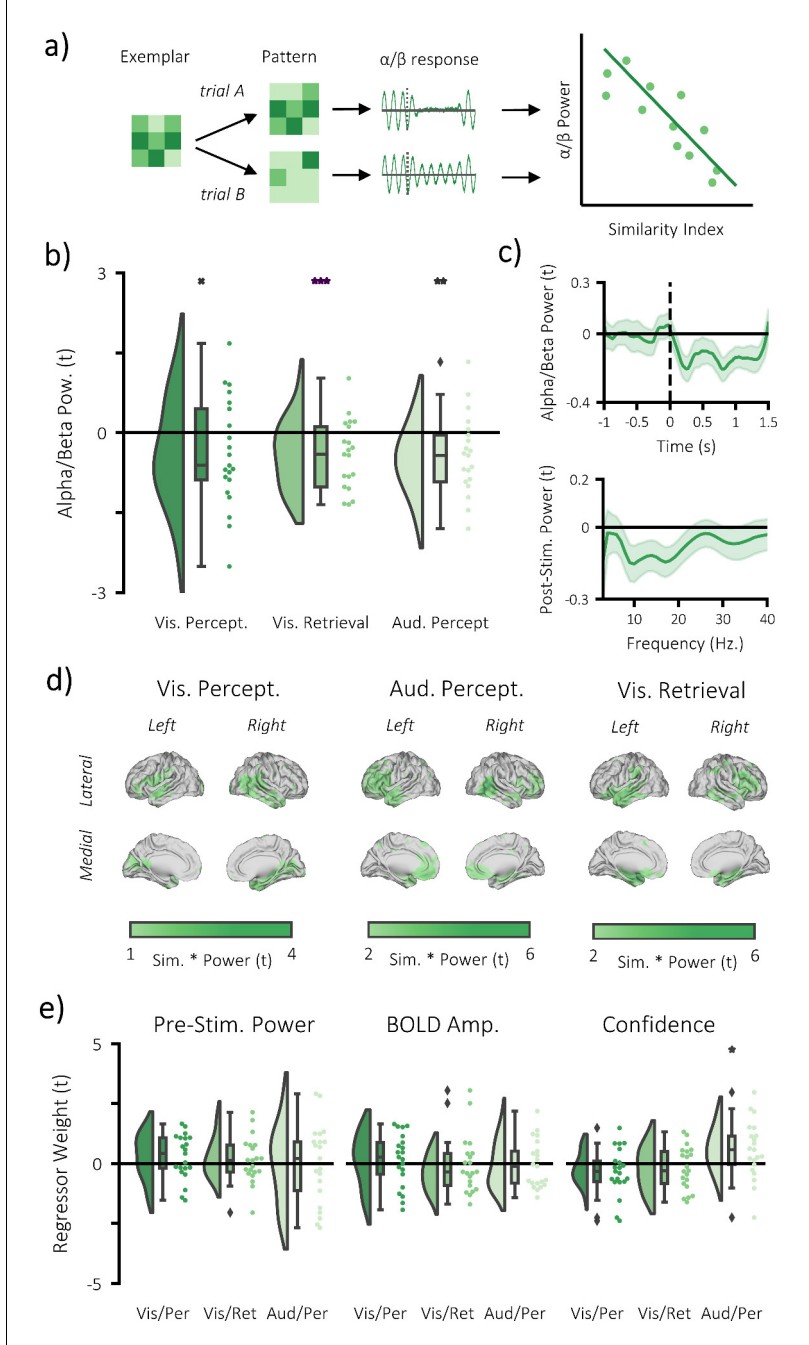

**Figure 4.** Alpha/beta power decreases track the fidelity of stimulus-specific information. (**a**) infographic depicting hypotheses and analytical approach. We anticipated that the more a pattern represented matching stimuli relative to differing stimuli, the greater the post-stimulus decrease in alpha/beta power would be. (**b**) Raincloud plot displaying the correlation between alpha/beta power and stimulus-specific information during visual perception, visual memory retrieval and auditory perception (each dot represents a single participant; ˣp = 0.054, \**p<0.01, \***p<0.001). (**c**) temporal (top) and spectral (bottom) specificity of the correlation between stimulus-specific information and alpha/beta power across all tasks (see *Figure 4—figure supplement 2* for each task individually). A value below zero indicates a negative correlation between variables. The negative relationship becomes apparent after stimulus onset (time = 0) within the frequency range 8–30 Hz. (**d**) brain map of the correlation between alpha/beta power at each virtual electrode with the measure of stimulus-specific information during visual perception, auditory perception and visual retrieval. (**e**) Raincloud plot displaying the correlation between stimulus-specific information and additional regressors 'pre-stimulus alpha/beta power', 'BOLD amplitude', and 'confidence rating' for each task (each dot represents a single participant; \*p<0.05).
*Figure 4 continued on next page*

*Figure 4 continued*

The online version of this article includes the following figure supplement(s) for figure 4:

**Figure supplement 1.** Correlation between alpha/beta power and BOLD signal during visual memory retrieval.
**Figure supplement 2.** Specificity of power-similarity correlation across time (top) and frequency (bottom).
**Figure supplement 3.** Investigating bimodal alpha power and stimulus-specific information.
**Figure supplement 4.** Separate contributions of periodic and aperiodic signal to event-related decreases in power.
**Figure supplement 5.** The simulated impact of epoch duration on the estimate of the 1/f curve.
**Figure supplement 6.** The simulated impact of measure variability and noise on the correlation between two variables.

correlated with post-stimulus alpha/beta power in a multiple regression for each participant (see *Figure 4a*), which included two additional regressors that had been shown to correlate with alpha/beta power (BOLD amplitude and confidence rating; see *Figure 4—figure supplement 1*/ Materials and methods) and a regressor to account for changes in pre-stimulus power (for example *van Dijk et al., 2008*; *Wöstmann et al., 2019*; *Iemi et al., 2017*). The resulting beta weights were transformed into t-statistics to standardise the measurement across regressors and participants. These t-statistics were then contrasted against the null hypothesis (there is no correlation; t = 0) in a one-sample t-test across participants. During visual perception, we found a trending negative correlation (p=0.054, Cohen's $d_z$ = 0.34), where a reduction in alpha/beta power was accompanied by an increase in stimulus-specific information (see *Figure 4b*). The additional regressors did not correlate with stimulus-specific information (see *Figure 4e*; BOLD amplitude: p=0.917, Cohen's $d_z$ = 0.02; confidence rating: p=0.736, Cohen's $d_z$ = 0.07; pre-stimulus alpha/beta power: p=0.141, Cohen's $d_z$ = 0.34). During auditory perception, we found a larger, significant negative correlation (p=0.006, Cohen's $d_z$ = 0.56), where a reduction in alpha/beta power was accompanied by an increase in stimulus-specific information (see *Figure 4b*). Neither BOLD amplitude (p=0.311, Cohen's $d_z$ = 0.23) nor pre-stimulus power (p=0.123, Cohen's $d_z$ = 0.34), correlated with the stimulus-specific information, though confidence rating did (p=0.031, Cohen's $d_z$ = 0.47). These results indicate that post-stimulus alpha/beta power decreases correlate with an increase in perceived stimulus-specific information.

We then aimed to replicate this effect in the retrieval task, working on the assumption that if alpha/beta power decreases are a proxy for information processing, the phenomenon should generalise across cognitive tasks. The correlation analysis was restricted to remembered trials to avoid a spurious correlation driven by memory-related differences in the decreases of alpha/beta power and increases of stimulus-specific information for remembered compared to forgotten trials. Representational distance was calculated between each single trial at retrieval and all trials at perception within the region of interest (i.e. the significant clusters identified in the fMRI searchlight analysis; see *Figure 2d*), and then correlated with a model that stated that representational distance for perceived stimuli matching the retrieved stimulus on this trial would be zero and representational distance for perceived stimuli differing from the retrieved stimulus on this trial would be one. The remainder of the analysis is the same as described above. In line with the previous results, we found a significant negative correlation for remembered trials (p<0.001, Cohen's $d_z$ = 0.56), where a reduction in post-stimulus alpha/beta power was accompanied by an increase in stimulus-specific information (see *Figure 4b–d*). The additional regressors did not correlate with stimulus-specific information (BOLD amplitude: p=0.598, Cohen's $d_z$ = 0.12; confidence rating: p=0.798, Cohen's $d_z$ = 0.06; pre-stimulus alpha/beta power: p=0.204, Cohen's $d_z$ = 0.30).

It is worth noting that a bimodal distribution of power could explain these results (*Freyer et al., 2009*). To address this, we re-ran our regression analyses, this time using a median split to divide trials based on whether they had high or low alpha/beta power (approximating a bimodal split of data) and entering this binary regressor into the regression model in place of the continuous alpha/beta power regressor used previously. This approach found a trending link between stimulus-specific information and alpha/beta power during visual memory retrieval (p=0.056, Cohen's $d_z$ = 0.37; see *Figure 4—figure supplement 3*), but not during visual perception (p=0.317, Cohen's $d_z$ = 0.12) or auditory perception (p=0.222, Cohen's $d_z$ = 0.18), suggesting that this effect cannot be consistently explained by a bimodal distribution of alpha/beta power. Furthermore, visual inspection of the

distribution of alpha/beta power within each participant suggests that alpha/beta power is normally distributed around the mean, rather than bi-modally distributed (see *Figure 4—figure supplement 3*).

It is also worth considering that functionally relevant changes in alpha/beta power needn't necessarily reflect a change in oscillatory power, but rather a change in the fractal signal (*Haller, 2018*; *Miller et al., 2009*) (often referred to as the 1/f curve). In line with these ideas, we found that event-related decreases in power are a summation of decreases in oscillatory alpha/beta power and changes in the shape of the 1/f curve (see *Figure 4—figure supplement 4*). Interestingly however, memory-related power changes were only reflected in decreases in alpha/beta oscillatory signal. Unfortunately, we were unable to link these measures to stimulus-specific information. Quite possibly, this shortcoming is due to the fact that the 1/f curve can only be properly observed (and hence analysed) over long epochs (i.e. several seconds to several minutes; *Miller et al., 2009*), or when the power-spectra are averaged across trials. Indeed, we demonstrate this principle using simulated data (see *Figure 4—figure supplement 5*). Short epochs (<5 s) give unreliable estimates of the 1/f curve whereas longer epochs (>30 s) appear much more robust. As our central analysis was event-related and conducted on the single trial level, any approach we take would be curtailed by the biologically infeasibility of getting a robust estimate of the 1/f curve in such a brief time window.

In sum, these results suggest that alpha/beta power parametrically decreases as the amount of stimulus-specific information represented within the cortex increases. While the chance of finding such an effect in the visual perception condition was marginal ($p=0.054$), it is worth noting that its conceptual equivalents yielded more substantial support (auditory perception: $p=0.006$; visual retrieval; $p<0.001$). Taking all these results together, there is substantial support to suggest that post-stimulus alpha/beta power decreases correlate with stimulus-specific information.

## Alpha/beta power decreases do not represent perceived or retrieved information

Lastly, we asked whether the observed negative correlation between alpha/beta power and stimulus-specific information could be explained by the fact that alpha/beta power, rather than providing favourable conditions for the brain to represent activity, actually represents information itself. To test this hypothesis, we conducted spatiotemporal representational similarity analysis (i.e. across virtual electrodes, time windows [500 to 1500 ms, in steps of 100 ms] and frequency bins [8 to 30 Hz, in steps of 1 Hz]) within the regions where stimulus-specific information was identified in the BOLD signal during perception and memory retrieval. By restricting analysis to regions where we had previously detected stimulus-specific information, we maximise our chance of finding an effect. Despite this extremely liberal approach, a cluster-based, permutation t-test found no evidence to suggest that alpha/beta power represents stimulus-specific information during visual perception ($p=0.548$, Cohen's $d_z = 0.03$), auditory perception ($p=0.773$, Cohen's $d_z = 0.17$) or visual retrieval ($p=0.579$, Cohen's $d_z = 0.04$). Notably, the frequentist nature of this test means we cannot conclude that alpha/beta power does not represent information, but rather that there is insufficient evidence to conclude that alpha/beta power represents information. To address this limitation, we ran a Bayesian one-sample t-test to probe the nature of the evidence in favour of the null hypothesis. The continued use of the region of interest switches the test from a liberal test of the alternative hypothesis (alpha/beta represents information) to a conservative test of the null hypothesis (alpha/beta does not represent information). Bayesian one-sample t-tests revealed moderate evidence in favour of the null hypothesis for the visual perceptual ($BF_{10} = 0.230$), auditory perceptual ($BF_{10} = 0.272$), and visual retrieval tasks ($BF_{10} = 0.232$). These results suggest that the observed relationship between alpha/beta power decreases and stimulus-specific information cannot be explained by the hypothesis that alpha/beta power itself represents information. Rather these results suggest that alpha/beta power decreases are a marker of the fidelity of stimulus-specific information.

## Discussion

Here, we provide empirical evidence to suggest that task-induced alpha/beta power decreases track the fidelity of stimulus-specific information represented within the cortex. We correlated simultaneously recorded alpha/beta power (as measured using scalp EEG) with a metric of stimulus-specific information (as quantified using representational similarity analysis [RSA] on fMRI data) on a trial-by-

trial level. As stimulus-specific information increased, alpha/beta power decreased, regardless of whether the information was externally presented or internally generated, or whether the information was visual or auditory. Further analysis revealed that this effect is not driven by the fact that alpha/beta power decreases represent information, suggesting instead that these decreases provide conditions which are beneficial for information processing.

Our central finding demonstrates that as alpha/beta power decreases, the fidelity of stimulus-specific information within the cortex increases. Task-related decreases in alpha/beta power are observable across tasks (*Pfurtscheller et al., 1994*; *Krause et al., 1994*; *Crone et al., 1998*; *Griffiths et al., 2016*; *Hanslmayr et al., 2009*; *Waldhauser et al., 2016*; *Obleser and Weisz, 2012*), sensory modalities (*Pfurtscheller et al., 1994*; *Krause et al., 1994*; *Crone et al., 1998*), and species (*Pfurtscheller et al., 1994*; *Haegens et al., 2011*; *Wiest and Nicolelis, 2003*; *Chatila et al., 1992*). Given their ubiquity, it stands to reason that they reflect a highly general cognitive process. Our results suggest that these alpha/beta power decreases are a proxy for information processing. Mechanistically speaking, these decreases may mark information processing as they allow for a reduction of neuronal noise correlations (which map onto local field potential; LFP; *Cui et al., 2016*). Numerous studies have demonstrated that task-irrelevant correlated activity between pairs of neurons is detrimental to stimulus processing (*Harris and Thiele, 2011*; *Zohary et al., 1994*) – particularly for large networks of correlated neurons (*Averbeck et al., 2006*) that, incidentally, are more likely to be detected in the LFP. Following the hypothesis that alpha/beta power decreases are a proxy for reductions in noise correlations (as opposed to a proxy for increased signal representation), one would predict that alpha/beta power decreases do not carry representational information about a stimulus. Rather, they provide favourable (i.e. reduced noise) conditions in which another mechanism can allow the internal representation of said stimulus to come forth. In line with this hypothesis, we found moderate evidence to suggest that alpha/beta power decreases do not carry any stimulus-specific information during the perception or retrieval of the visual stimuli. Notably, this does not contradict previous findings showing that stimulus specific information is coded in the phase of alpha oscillations (*Michelmann et al., 2016*). As the power and phase of an oscillation are mathematically independent, it is entirely plausible to suggest that one carries stimulus-specific information while the other does not. As such, one could view alpha/beta power decreases as a marker for the potential for information processing, rather than representing information.

Alpha/beta activity has also been linked to information representation based on the ideas of information theory (*Hanslmayr et al., 2012*; *Shannon and Weaver, 1949*). Information theory proposes that little information can be gathered from a highly predictable input (e.g. a network of highly correlated, spiking neurons) – if you can predict an upcoming event, you must already know details about the event. In contrast, a lot of information can be gathered from unpredictable inputs (e.g. uncorrelated spiking neurons) – you learn a lot from a completely novel experience. It has been theorised that desynchronisation within the alpha and beta bands reduces the predictability of neuronal firing and hence boosts information processing abilities (*Hanslmayr et al., 2012*). For example, earlier work has demonstrated that tasks which involve greater semantic elaboration (i.e. greater information processing) produce greater alpha/beta power decreases (*Hanslmayr et al., 2009*). Our central result fits neatly within this framework as we find that alpha/beta power decreases parametrically increase with the presence of stimulus-specific information. Moreover, our finding that alpha/beta power does not directly represent stimulus-specific information fits with this idea, as these power decreases are theorised to allow complex neuronal patterns to emerge rather than generate the complex patterns themselves. Taken together, one could speculate that alpha/beta power decreases allow for the rich representation of stimulus-specific information by reducing the predictability of neural firing patterns. Notably, the information theoretic interpretation (i.e. predictable firing is bad for information processing) is highly similar to the idea that correlated firing (i.e. noise correlations) is bad for information processing because correlations are inherently predictable. This opens an exciting new line of investigation which would aim to tease apart these two hypotheses. One could directly compare whether the indiscriminate attenuation of all synchronised neuronal firing (i.e. the reduction of any form of neural synchrony) better benefits information representation than the selective attenuation of neurons that contribute to noise correlations (i.e. only the reduction of task-irrelevant neural synchrony). Evidence supporting the former would suggest that information theory would be a better framework for understanding cortical information representation, while

evidence supporting the latter would suggest that noise correlations be describe cortical information representation.

Several established accounts have interpreted high-amplitude alpha oscillations as a marker for inhibition (*Pfurtscheller et al., 1996*; *Jensen and Mazaheri, 2010*; *Klimesch et al., 2007*). One may wonder, then, how the current results can be reconciled with these established accounts. Quite simply, we view the information processing account and the existing inhibition accounts as two sides of the same coin. Earlier accounts focus upon how alpha power increases reflect inhibition, our framework focuses on the complementary idea that alpha power decreases boost information representation through disinhibited networks. Importantly, we expand on these earlier accounts by demonstrating that alpha/beta power does not simply reflect a binary division between inhibition and disinhibition. Rather, alpha/beta power can parametrically track the degree to which a network can represent information. In other words, as alpha/beta power gets progressively weaker, the network becomes progressively disinhibited and, therefore, more capable of establishing detailed neural representations.

It is worth noting that we did not observe a correlation between stimulus-specific information and pre-stimulus alpha/beta power. This presents an apparent contradiction to earlier work which has shown that a decrease in pre-stimulus alpha power correlates with an increase in perceptual performance (*van Dijk et al., 2008*; *Hanslmayr et al., 2007*). A potential resolution to this peculiarity lies in a number of recent studies (*Wöstmann et al., 2019*; *Iemi et al., 2017*; *Lange et al., 2013*; *Samaha et al., 2017*; *Limbach and Corballis, 2016*; *Benwell et al., 2017*; *Whitmarsh et al., 2017*) which have used signal detection theory (*Green and Swets, 1966*) to disentangle objective and subjective measures of perceptual performance (termed 'sensitivity' and 'decision criterion' respectively). These studies demonstrate that the link between pre-stimulus alpha power and perceptual performance can be better explained by decision criterion (e.g. increases in responses rates; confidence; awareness) rather than by sensitivity (e.g. increased task accuracy). Under this framework, we view our measure of stimulus-specific information as one of sensitivity, as it reflects the veridical representation of information within the brain rather than the subjective experience of this information. As these previous studies have demonstrated consistently that pre-stimulus power does not correlate with measures of sensitivity, it is perhaps no surprise that we found no correlation between stimulus-specific information and pre-stimulus power.

Recent work has begun to emphasise the importance of distinguishing periodic neural activity (which can approximate oscillatory activity) from aperiodic activity (which reflects the 1/f power law) (*Haller, 2018*; *Miller et al., 2009*). We attempted to address this question in our EEG and combined EEG-fMRI analyses with partial success. In our EEG analyses, we demonstrated that stimulus-induced reductions in alpha/beta power are a summation of decreases in alpha/beta oscillatory activity, a flattening of the 1/f curve, and an overall increase in power. These results suggest that stimulus-induced decreases in alpha/beta power are more complicated than a simple reduction in oscillatory power. Intriguingly, memory-related change in alpha/beta activity could only be explained by a decrease in alpha/beta oscillatory power (confirming earlier findings which suggest that the 1/f curve cannot explain memory-related changes in power; *Fellner et al., 2019*). The contrast between these two tasks sheds light on what periodic and aperiodic measures of power may be reflecting on a cognitive level. Throughout this paper, we have emphasised how information representation is a task-general phenomenon; the fact that changes in slope only appear in post-stimulus vs. pre-stimulus perceptual contrasts and not in remembered vs. forgotten memory contrasts suggests that a change in slope does not meet the task-general requirement of the hypothesis. However, the presence of alpha/beta oscillatory power decreases in both analyses means that changes in oscillatory activity may still relate to information representation. Unfortunately, we were unable to confirm this idea in the combined EEG-fMRI analyses. We would have expected stimulus-specific information to correlate with oscillatory power decreases but not changes in the slope. Our analyses, however, returned inconclusive evidence where no EEG measure appeared to correlate with stimulus-specific information. It is unclear why this occurred, but one plausible explanation is that the 1/f curve cannot be properly estimated for single-trial, event-related analyses. As the 1/f slope is a product of many seconds of recorded signal, it cannot be estimated 'instantaneously' (*Miller et al., 2009*). Supplementary simulations (see *Figure 4—figure supplement 5*) demonstrate this. Short epochs (<5 s) give unreliable estimates of the 1/f curve, but these estimates stabilise as epoch length increases (>20 s) or when epochs are averaged together. As our trial epochs are 1 s long, it seems implausible to suggest that

any reliable separation of the 1/f curve and oscillatory power can be computed here. Therefore, any result from this analysis should be treated with caution, with inferences best left to the condition-based analyses of oscillatory alpha/beta power, which show that their decreases transcend both stimulus modality and task – fitting a domain-general information processing mechanism.

Intriguingly, our analysis revealed that the correlation between alpha/beta power decreases and stimulus-specific information was weakest during visual perception. Given that stimuli in the auditory perception task had to compete with the MRI scanner noise (potentially resulting in degraded neural representations) and the stimuli in the visual retrieval task may have been subject to retroactive inter-ference, one would have anticipated that the measures of stimulus-specific information would have been most consistent in the visual perception task. In fact, this may be the very reason why the correlation was weakest in the visual perception task. Working on the assumption that alpha/beta power does correlate with stimulus-specific information, then in a task where neural representations of stimuli are near-perfect on every trial, alpha/beta power should not be expected to fluctuate greatly across trials either. If a small amount of noise is injected into either measure, then the correlation will be greatly reduced. If, however, neural representations of stimulus are highly variable across trials and alpha/beta power co-varies with these fluctuations, then a small amount of noise would have a less substantial impact on the correlation between stimulus-specific information and alpha/beta power. Simulations in *Figure 4—figure supplement 6* demonstrate this principle. In short, the comparatively small effect in the visual perception task relative to the auditory perception and memory retrieval tasks may be explained by a limited variation in stimulus representation across trials.

Both the causality and directionality of the central result remain open to debate. Perhaps the most critical question is whether alpha/beta power decreases are a prerequisite for information proc-essing. We speculate that this is not the case. Our theoretical interpretation of the results views these power decreases as a means to boost a stimulus's signal-to-noise ratio by reducing noise cor-relations. Arguably however, the stimulus's signal-to-noise ratio can also be boosted by increasing the stimulus's signal intensity (*Fries, 2015*). This would lead us to hypothesise that alpha/beta power decreases are sufficient, though not necessary, for information processing. This hypothesis would explain the size of the per-subject correlation values observed here and in previous studies that linked noise correlations and information processing (*Zohary et al., 1994*; *Cohen and Kohn, 2011*) – if other processes contribute to information processing, the correlation will not be perfect. Indeed, this hypothesis is supported by a study where task-related alpha/beta power decreases were dis-rupted by transcranial magnetic stimulation (*Waldhauser et al., 2016*) (TMS). In this study, TMS reduced behavioural performance (suggesting that task-related alpha/beta power decreases facili-tate information processing), but did not render participants completely incapable of recalling infor-mation (suggesting other processes also contribute to information processing). This reasoning generates an interesting question: does brain stimulation impair measures of stimulus-specific infor-mation in the BOLD signal by entraining alpha/beta activity? Addressing this question would help to clarify the extent to which alpha/beta activity influences the representation of stimulus-specific infor-mation within the cortex.

In this experiment, we focused on the alpha/beta frequencies (8–30 Hz) for both theoretical (*Hanslmayr et al., 2012*) and pragmatic reasons (*Fellner et al., 2016*). This focus does ask however: do the theta and gamma frequencies (3–7 Hz; 40–100 Hz) relate to information in a similar manner? Both the perception and retrieval of stimuli typically induce power increases in the theta and gamma bands (for example *Jutras et al., 2009*). These power increases are not overtly congruent with the theories of information processing via neuronal decoupling (*Harris and Thiele, 2011*; *Averbeck et al., 2006*; *Zohary et al., 1994*) or neuronal unpredictability (*Hanslmayr et al., 2012*). As alpha/beta power decreases are proposed to facilitate information processing by reducing noise, however, these theta or gamma power increases could theoretically facilitate information processing through the complementary means of increasing signal strength. For example, the 'communication through coherence' hypothesis proposes that neuronal representations of a stimulus are enhanced by an increase in gamma synchronicity (*Fries, 2015*). Given that alpha/beta power decreases fre-quently co-occur with gamma power increases (for example *Burke et al., 2014*), one could speculate that these two mechanisms interact such that the former reduces noise while the latter boosts signal to further optimise the efficiency of information processing.

In conclusion, we find evidence to suggest that alpha/beta power decreases track the fidelity of stimulus-specific information represented within the cortex. Given that these alpha/beta power decreases are observed across tasks (*Pfurtscheller et al., 1994*; *Krause et al., 1994*; *Crone et al., 1998*; *Griffiths et al., 2016*; *Hanslmayr et al., 2009*; *Waldhauser et al., 2016*; *Obleser and Weisz, 2012*), sensory modalities (*Pfurtscheller et al., 1994*; *Krause et al., 1994*; *Crone et al., 1998*), and species (*Pfurtscheller et al., 1994*; *Haegens et al., 2011*; *Wiest and Nicolelis, 2003*; *Chatila et al., 1992*), it stands to reason that they reflect a highly general cognitive process. Our findings suggest that such power decreases reflect enhanced information processing. These power decreases may act as a proxy for information processing either through their link to reduced neuronal noise correlations (*Harris and Thiele, 2011*; *Averbeck et al., 2006*; *Cui et al., 2016*) or by reducing the predictability of neuronal activity (*Hanslmayr et al., 2012*). These results open numerous avenues for future research, such as how these decreases interact with other neural processes to facilitate the representation of stimulus-specific information, and whether brain stimulation can be used to manipulate the fidelity of information represented within the cortex. Ultimately, these results further illuminate how the ubiquitous phenomenon of task-related alpha/beta power decreases relate to the processing and comprehending of our physical and mental worlds.

## Materials and methods

### Participants

Thirty-three participants were recruited. All participants were Native English speakers with normal or corrected-to-normal vision. In return for their participation, they received course credit or financial reimbursement. Twelve of these participants were excluded from analysis: one participant was excluded due to recording issues relating to the MRI scanner, three participants were excluded due to recording issues relating to the EEG system, five participants had insufficient recalled pairs (n < 10) following EEG artifact rejection, and three participants had insufficient forgotten pairs (n < 10) following EEG artifact rejection. This left twenty-one participants for statistical analysis, approximately matching the sample size used by Michelmann and colleagues when they used this paradigm (*Michelmann et al., 2016*). Ethical approval was granted by the Research Ethics Committee at the University of Birmingham, complying with the Declaration of Helsinki.

### Behavioural paradigm

Each participant completed a paired associates task (*Michelmann et al., 2016*; *Griffiths et al., 2019*) (see *Figure 1b*). During encoding, participants were presented with a 3 s video or sound, followed by a noun. There was a total of four videos and four sounds, repeated throughout each block. All four videos had a focus on scenery that had a temporal dynamic, while the four sounds were melodies performed on four distinct musical instruments. Participants were asked to 'vividly associate' a link between every dynamic and verbal stimulus pairing. For each pairing, participants were asked to rate how plausible (one for very implausible and four for very plausible) the association they created was between the two stimuli (the plausibility judgement was used to keep participants on task rather than to yield a meaningful metric, and to ensure that motion in perceptual and retrieval blocks was consistent to aid comparability between tasks). The following trial began immediately after participants provided a judgement. If a judgement was not recorded within 4 s, the next trial began. This stopped participants from elaborating further on imagined association they had just created. After encoding, participants completed a 2 min distractor task which involved making odd/even judgements for random integers ranging from 1 to 99. Feedback was given after every trial. During retrieval, participants were presented with every word that was presented in the earlier encoding stage and, 3 s later, asked to identify the associated video/sound from a list of all four videos/sounds shown during the previous encoding block. The order in which the four videos/sounds were presented was randomised across trials to avoid any stimulus-specific preparatory motor signals contaminating the epoch. Following selection, participants were asked to rate how confident they felt about their choice (one for guess and four for certain). Each block consisted solely of video-word pairs or solely of sound-word pairs – there were no multimodal blocks. Each block consisted of 48 pairs, with each dynamic stimulus being presented an equal number of times (i.e. 12 repetitions of each dynamic stimulus). There were four blocks in total. After the second block, the structural T1-

weight image was acquired, giving participants a chance to rest. Any participant that had fewer than 10 'remembered' or 10 'forgotten' trials after EEG pre-processing were excluded from further analysis. All participants completed the task in the MRI scanner, with fMRI and EEG data acquisition occurring at both encoding and retrieval. Responses were logged using NATA response boxes.

### Behavioural analysis

Trials were characterised as 'remembered' or 'forgotten'. Remembered trials corresponded to those in which the participant could link the verbal cue to the correct video, and indicated that their decision was not a guess (i.e. confidence rating >1). Forgotten trials corresponded to those in which the participant could not link the verbal cue to the correct video, or indicated that their decision was a guess (i.e. confidence rating = 1). While earlier studies using this paradigm (*Michelmann et al., 2016*; *Griffiths et al., 2019*) have only considered 'highly confident' memories (i.e. max confidence rating), we chose a more lax confidence threshold to ensure that sufficient trials of each dynamic stimulus available for the fMRI representational similarity analysis. Under these criteria, participants (on average) correctly recalled 63.4% of the video-word pairs (s.d. 7.5%; range: 47.6–74.5%).

### fMRI acquisition

The magnetic resonance imaging data were acquired using a 3T Philips scanner with a 32-channel SENSE receiver coil at the Birmingham University Imaging Centre (BUIC). Participants were instructed to avoid moving as much as they could, and motion was further restricted by placing foam pads inside the RF coil. Functional volumes consisted of 32 axial slices (4 mm thickness) with 3 $\times$ 3 mm voxels, providing full head coverage (field of view: 192 $\times$ 192$\times$128 mm), acquired through an echo-planar imaging (EPI) pulse sequence (TR = 2 s, TE = 40 ms, flip angle of 80°). Four dummy scans were acquired immediately prior to the beginning of each run to allow for magnetic field stabilisation. Eight runs were obtained (four encoding runs and four retrieval runs), each of which acquired 255 volumes plus four dummy scans. A T1-weighted structural image (1 $\times$ 1$\times$1 mm voxels; TR = 7.4 ms; TE = 3.5 ms; flip angle = 7°, field of view = 256$\times$256 x 176 mm) was acquired after the second block.

### fMRI pre-processing

Pre-processing of the fMRI data was conducted in SPM 12. The functional images first underwent slice time correction, followed by spatial realignment to the first volume of each run. The structural T1-weighted image was then co-registered to the mean image of the functional MRI data. The co-registered T1-weighted image was then segmented. For the univariate analysis, the functional and structural images were normalised to MNI space, and then smoothed using a 8 $\times$ 8$\times$8 mm full-width at half-maximum (FWHM) Gaussian kernel. For the RSA analyses, the data were kept in native space and not smoothed as this approach is optimal for searchlight analysis (*Limbach and Corballis, 2016*).

### fMRI representational similarity analysis

Searchlight-based representational similarity analysis (RSA) was conducted using a combination of the MRC CBU RSA toolbox (http://www.mrc-cbu.cam.ac.uk/methods-and-resources/toolboxes/) and custom scripts (https://github.com/benjaminGriffiths/reinstatement_fidelity; copy archived at https://github.com/elifesciences-publications/reinstatement_fidelity). Representational distance was quantified as the cross-validated Mahalanobis (CVM) distance (*Nili et al., 2014*; *Walther et al., 2016*), which provides an unbiased measure of pattern dissimilarity (*Walther et al., 2016*). The CVM approach takes a training dataset and finds weights that maximises the Euclidean distance between two stimuli. These weights are then applied to a testing dataset, and the weighted Euclidean (i.e. cross-validated Mahalanobis) distance is calculated between stimuli. For analysis of both tasks, the covariance was estimated on the training data across all four categories (that is, it is an estimate of the noise covariance). As this covariance may be rank deficient (*Walther et al., 2016*), the matrix underwent shrinking towards the diagonal matrix using the optimal shrinkage factor as described by *Ledoit and Wolf (2004)*. For the analysis of the perceptual task, the time-corrected and spatially-realigned fMRI data were demeaned and then split into two partitions, with the first partition containing data from the first block and the second partition containing data from the second block. A

general linear model (GLM) was then used to estimate the BOLD response for each category, separately for the two partitions (*Nili et al., 2014*; *Walther et al., 2016*). Four regressors of interest were included (that is, one regressor for each video). For each of these regressors, each video onset was modelled as a stick function spanning the duration of the video, which was then convolved with a canonical hemodynamic response function (HRF). The first partition served as training data for calculating CVM distance on the second partition, and the second partition served as training data for calculating distance on the first partition. CVM distance was computed between every stimulus pattern at encoding. The derived CVM distance was then correlated with a hypothesised model, which stated that (i) there would be a perfect correlation ($r = 1$) between the representation of each repetition of the same video, and (ii) there would be no correlation ($r = 0$) between the representation of differing videos. Spearman's correlation was used based on the ordinal nature of the hypothesised model. The resulting correlation co-efficient was then corrected using the Fisher z-transform to approximate a normal distribution. This analysis was conducted across the whole brain using searchlights with a radius of 10 mm (i.e. 121 voxels). Searchlights that contained less than 60% of these 121 voxels (e.g. searchlights in the most lateral areas of the neocortex) were discarded from analysis. The Fisher z-value of each searchlight was placed in a brain map, at the centre voxel of the searchlight. For statistical inference, the resulting brain maps of each subject were analysed in a second-level one-sample t-test. The resulting group-level whole-brain map was thresholded in SPM using $p_{uncorr.}$ <0.001 and a cluster extent of k = 10. Clusters that were formed using this threshold were then considered 'significant' if the cluster-level $p_{FWE}$ value was less than 0.05. Notably, such cluster-forming thresholding may inflate from family-wise error rate from the expected 5% to 7–10% (*Eklund et al., 2016*) and hence classify marginal effects as erroneously significant. As such, we cross-checked our results with a more conservative cluster-forming threshold of $p_{uncorr}$ <0.0001 and a cluster extent of k = 50 (see *Supplementary file 1*, Supplementary table 1). This stricter threshold did not influence the central results.

For the retrieval task, this analysis was adapted slightly. The cross-validation method used above assumes that each representation of the same video is identical, and while this is true for perception (participants always viewed one of the four identical video clips), the same is not true for retrieval (each memory consists of a unique word-video pair). To address this concern, trials that contained the same video were averaged together to maximise the video-stimulus 'signal' and minimise the word-stimulus 'noise'. These mean patterns were then subjected to the same analysis as above. Weights maximising the Euclidean distance between each mean pattern were calculated on a training dataset, and applied to the testing dataset to allow the calculation of the CVM distance. This was conducted between every pattern at both perception and retrieval. The observed distances were then correlated with a hypothesised model, which stated that (i) there would be a perfect correlation ($r = 1$) between the mean representation of a video at retrieval and the mean representation of the same video at perception, and (ii) there would be no correlation ($r = 0$) between the mean representation of a video at retrieval and the mean representations of differing videos at perception. Any cases of perception-perception or retrieval-retrieval similarity were excluded from this model, meaning this model isolates the effects of memory reinstatement. As with the perceptual RSA analysis, the retrieval RSA analysis was conducted across the entire brain, and comparisons were corrected for accordingly. The approaches to searchlight analysis and statistical inference were identical to those described in the previous paragraph.

## EEG acquisition

The EEG was recorded using a MR compatible Brain Products system (Brain Products, Munich, Germany) and a 64-electrode cap with a custom layout (including an EOG and ECG channel). As movement within the scanner has been shown to profoundly impair EEG data quality (*Fellner et al., 2016*), motion sensors were attached to the EEG cap to assist in the attenuation of movement-related EEG artifacts (*Jorge et al., 2015*). Briefly, this method involves placing plastic tape under four electrodes (10–10 positions F5, F6, T7 and T8) to insulate these electrodes from the scalp, then adding an external wire to complete the circuit between the channel and the reference. Consequently, the activity recorded on these channels is the product of changes in magnetic flux. The EEG sampling rate was set to 5 kHz. Impedances were kept below 20 kΩ. All electrode positions, together with the nasion and left and right pre-auricular areas were digitised using a Polhemus

Fasttrack system (Polhemus, Colchester, VT) for use in the creation of headmodels for source localisation.

## EEG preprocessing

All EEG analysis was carried out using MATLAB (MathWorks, Natwick, MA), the Fieldtrip (*Oostenveld et al., 2011*) and *fmrib* (*Niazy et al., 2005*; *Iannetti et al., 2005*) toolboxes, and custom scripts. The raw data were first high-pass filtered (1 Hz; FIR). Following this, the gradient artifact was corrected using the FASTR algorithm implemented in the *fmrib* toolbox (*Niazy et al., 2005*; *Iannetti et al., 2005*). The gradient template for each TR was modelled on the average gradient artifact of the 60 nearest TRs. Residual artifacts from the acquisition of each slice (32 slices in 2 s/16 Hz) were filtered out using a bandstop(15.5 Hz to 16.5 Hz) Butterworth filter. The data were then downsampled to 500 Hz and the ballistocardiogram (BCG) artifact was corrected using optimal basis set, again implemented in the *fmrib* toolbox. Heartbeat onsets were taken from the MR scanner's physiological recordings. The continuous data were then inspected for large periods of movement which were marked and the associated MR scanner triggers deleted. Subsequently, the gradient and BCG corrections were repeated on the continuous data with the periods of movement excluded. This helped improve the accuracy of the gradient and BCG templates that were subtracted from the data. After gradient and pulse artefact correction the data from the motion sensors were used in a multi-channel recursive least squares algorithm to regress out the remaining movement-related artifacts (*Bouchard and Quednau, 2000*; *Masterton et al., 2007*) (while retaining brain signal; *Daniel et al., 2019*) using custom scripts previously implemented by Jorge and colleagues (*Jorge et al., 2015*).

All subsequent EEG pre-processing was conducted using the Fieldtrip toolbox (*Oostenveld et al., 2011*). First, the data were epoched into trials beginning 2 s before the onset of the video at perception/cue at retrieval and ending 4 s after the onset of the cue. Second, independent component analysis was used to remove blinks, saccades and any residual spatially-stationary noise that appeared to be linked to the cardiac artifact. Third, the data were demeaned, low-pass filtered (100 Hz; Butterworth IIR) and re-referenced to the average of all channels. Fourth, the data were visually inspected to identify and reject any trials and/or channels containing residual artifacts (mean percentage of trials rejected: 23.1%; range: 10.4% to 39.1%). Fifth, the data were demeaned and re-referenced again to the average of all good channels (note that as any noise introduced by noisy channels in the earlier step will be shared by all good channels and therefore subtracted out during this re-referencing). Lastly, the scalp level data were reconstructed in source space to attenuate residual muscle artifacts (for details, see below).

## EEG source analysis

The preprocessed data were reconstructed in source space using individual head models, structural (T1-weighted) MRI scans and 4-layer boundary element models (BEM; using the dipoli method implemented in Fieldtrip). Electrode positions (as digitised via the Polhemus Fasttrack system) were mapped onto the surface of the scalp using fiducial points for reference. The timelocked EEG data were reconstructed using a Linearly Constrained Minimum Variance (LCMV) beamformer (*Van Veen et al., 1997*). The lambda regularisation parameter was set to 5%.

## EEG time-frequency analysis

First, the source-reconstructed EEG data were convolved with a 6-cycle wavelet (−1 to 3 s, in steps of 25 ms; 8 to 30 Hz; in steps of 0.5 Hz). Second, the resulting data were z-transformed using the mean and standard deviation of power across time and trials (*Griffiths et al., 2016*). Third, the data were restricted to two time/frequency windows of interest (−1000 to −375 ms and 500 to 1500 ms post-stimulus; both 8–30 Hz; *Michelmann et al., 2016*) and then averaged across these windows, resulting in two alpha/beta power values per trial for each virtual electrode. To probe whether alpha/beta power decreased following stimulus onset these two values were contrasted in a one-tailed, non-parametric, cluster-based permutation-based t-test (*Maris and Oostenveld, 2007*) with 2000 randomisations. To investigate whether alpha/beta power decreased for remembered relative to forgotten trials, the data for the post-stimulus window were split by condition and contrasted using the same statistical approach.

## Combined EEG-fMRI analysis

An adjusted CVM approach outlined in *fMRI representational similarity analysis* was used to quantify information for this analysis. Rather than use a searchlight, CVM distance was computed in a region of interest (ROI) defined by the searchlight analysis. Specifically, this ROI consisted of all voxels included in any significant cluster revealed in the earlier analysis plus all neighbouring voxels that would have been included in the searchlight that contributed to the cluster. This approach maximised signal-to-noise for the measure of stimulus-specific information by only focusing on voxels where stimulus-specific information could be detected (see below for a note on circularity). As before, a training dataset was used to find weights that maximally discriminates two stimuli (per trial for encoding; averaged across repetitions for retrieval). In the case of retrieval data however, rather than project these weights onto stimulus-averaged testing dataset, these weights were projected onto the trial-level dataset. These trial-level BOLD responses were estimated using a GLM where each trial was considered as a separate regressor (using a stick function convolved with a canonical HRF as in the earlier analyses). This change in approach provides a measure of stimulus-specific information for every trial within the specified ROI.

Similarly, an adjusted approach was used to quantify EEG power per trial. Whereas the prior section measured EEG power across all virtual electrodes, this analysis was restricted to virtual electrodes included in regions that coded for stimulus-specific information (as determined by the fMRI searchlight analysis). This approach ensured that the analysed EEG signal originated from the same region as the fMRI similarity index.

These approaches yield a single measure of fMRI-derived stimulus-specific information and EEG-derived alpha/beta power for every trial. A multiple regression was then conducted for each participant, with stimulus-specific information used as the outcome variable, and post-stimulus alpha/beta power (500–1500 ms) being used as the predictor. Three additional regressors were included to address potential confounds: confidence rating, BOLD amplitude within the ROI, and pre-stimulus alpha/beta power ($-1000$ to $-375$ ms). This returned an t-value for every regressor of every participant. Group-level statistical analysis saw these t-values being contrasted against the null hypothesis ($t = 0$; there is no correlation) in a one-tailed, non-parametric, permutation-based t-test (*Maris and Oostenveld, 2007*) with 2000 randomisations where the observed data and null hypothesis were permuted (again, separately for hits and misses in the case of the retrieval task). To test the spatial specificity of the effect, the correlation analyses above were re-run for each virtual electrode and then subjected to a one-tailed, non-parametric, cluster-based permutation-based t-test (*Maris and Oostenveld, 2007*).

We also addressed the spectral specificity of the effect. However, one should note that these results are difficult to interpret as both theta (3–7 Hz) and gamma (40–50 Hz) bands are much more susceptible to distortion by the MRI scanner than the alpha/beta band (*Fellner et al., 2016*). Aside from changes to the frequencies of interest, the analysis matched that which is described above. We considered both tails of the t-test, testing two differing hypotheses: 1) a reduction in power reflects an increase in information (mirroring the central hypothesis of the paper), and 2) as theta/gamma power typically increases during cognitive engagement (for example *Fries, 2015*), an increase in power reflects an increase in information. This effect did not generalise to the theta (visual perception p=0.347, visual memory retrieval p=0.486, auditory perception p=0.163) or gamma bands (perception p>0.5, visual memory retrieval p=0.153, auditory perception p<0.5).

## EEG-confidence correlation

It is plausible to suggest that the more information one recalls about an associated pair, the more confident they are in selecting the correct video. Therefore, it could be argued that alpha/beta power decreases are a confidence signal. To address this potential confound, we took our measure of EEG alpha/beta power and correlated this with the confidence rating provided on each trial. The derived r-value underwent Fisher z-transformation to approximate a normal distribution. These Fisher z-values were contrasted against the null hypothesis (there is no correlation; z = 0) across participants in a one-sample t-test. We found a significant negative correlation (p=0.033, Cohen's d = 0.48), where a reduction in alpha/beta power was accompanied by an increase in confidence rating. While these results indicate a link between alpha/beta power and confidence, this link does not

explain the link between alpha/beta power and stimulus-specific information (as evidenced by the partial correlation reported in the results section).

## EEG irregular-resampling auto-spectral analysis (IRASA)

IRASA analyses was conducted using the Matlab toolbox created by *Wen and Liu (2016)* (available at: https://purr.purdue.edu/publications/1987/1). The power spectra density (PSD) was estimated for each trial and then averaged across trials from the same condition (either post-stimulus power or pre-stimulus power for perceptual analyses; either remembered or forgotten power for the memory retrieval analysis) to get a robust estimate of the 1/f curve. The averaged PSD were then subjected to the IRASA algorithm, which splits the PSD into two power spectra – the fractal component (approximating the 1/f curve) and the oscillatory component (approximating underlying oscillatory activity). To get the slope and intercept of the 1/f curve, the fractal power spectrum (A) and its associated frequencies (B) were put into log-space (to provide a linear line), and then the linear equation A = Bx+y was solved using least-squares regression, where x is the slope of the 1/f curve and y is the intercept. Oscillatory power was calculated as the mean power between 8 and 25 Hz (as the maximum frequency to be derived is one quarter of the sampling rate [100 Hz]), matching the approach used for the wavelet analysis to facilitate cross-analysis comparison.

For the combined EEG-fMRI analysis, IRASA was computed on the single-trial level and then all three resulting estimates (slope, intercept, oscillatory power) were entered into a multiple regression in the same manner reported for the wavelet-based EEG-fMRI analyses. No effect was found for any element, perhaps because the 1/f cannot be robustly estimated over short time windows (*Miller et al., 2009*) (see *Figure 4—figure supplement 2*).

## A note on circularity

The use of data-driven regions of interest (ROIs) can, in some cases, introduce circularity into the analysis (*Kriegeskorte et al., 2009*). As a result, this can overestimate the size of an effect. However, we contend that our use of data-driven ROIs does not fall foul to this analytical flaw. Explicitly stated, the concern here is that by selecting the ROI that carries stimulus-specific information in the BOLD signal, we inflate the chance of finding a correlation between BOLD-derived stimulus-specific information and alpha/beta power in the same ROI. This concern is only valid when alpha/beta power also carries stimulus-specific information. In such an instance, we would essentially be limiting our correlation between two metrics of stimulus-specific information to a ROI where we know that (in this dataset) stimulus-specific information is represented. However, a Bayesian inference of RSA conducted on alpha/beta power (see results and section below) demonstrated that there is moderate evidence in favour of the null hypothesis that alpha/beta power does not carry stimulus-specific information. In light of this, we can infer that the use of data-driven ROIs in this instance does not introduce circularity into our analysis.

## EEG representational similarity analysis

To identify whether alpha/beta power carried stimulus-specific information, representational similarity analysis was conducted on the EEG time-frequency data (for perception and successful retrieval separately). The time-frequency data were derived in the same manner as described in the earlier section, but rather than average over time/frequency (as described in the third step), the individual time and frequency bins were retained. Representational similarity was quantified using Spearman's correlation across all features (i.e. time, frequency and location) of every pair of trials. The resulting value underwent Fisher-z transformation to approximate a normal distribution. The observed similarity was then contrasted against the same models used in the earlier RSA approaches. This resulted in a single value describing stimulus-specific information for each subject, which was tested against the null hypothesis (there is no stimulus-specific information in alpha/beta power) in a one-tailed, non-parametric, permutation-based t-test (*Maris and Oostenveld, 2007*).

As we found insignificant evidence to support the alternative hypothesis, we then took a Bayesian approach to the statistical analysis. The same values used in above were analysed in a Bayesian one-sample t-test (as implemented in JASP, version 0.9; *JASP-Team, 2018*). We interpreted the resulting Bayes factor in line with the rule of thumb (*Lee and Wagenmakers, 2013*).

## Additional information

### Funding

| Funder | Grant reference number | Author |
|---|---|---|
| H2020 European Research Council | 647954 | Simon Hanslmayr |

The funders had no role in study design, data collection and interpretation, or the decision to submit the work for publication.

### Author contributions

Benjamin James Griffiths, Data curation, Formal analysis, Visualization, Methodology; Stephen D Mayhew, Karen J Mullinger, Resources, Formal analysis, Supervision, Methodology; João Jorge, Resources, Methodology; Ian Charest, Maria Wimber, Formal analysis, Supervision; Simon Hanslmayr, Conceptualization, Supervision, Funding acquisition, Project administration

### Author ORCIDs

Benjamin James Griffiths (iD) https://orcid.org/0000-0001-8600-4480
Simon Hanslmayr (iD) https://orcid.org/0000-0003-4448-2147

### Ethics

Human subjects: Participants provided informed consent to the experiment, the publication of the results, and the uploading of their anonymised data. Ethical approval was granted by the Research Ethics Committee at the University of Birmingham (ERN_15-0335B), complying with the Declaration of Helsinki.

### Decision letter and Author response

Decision letter https://doi.org/10.7554/eLife.49562.sa1
Author response https://doi.org/10.7554/eLife.49562.sa2

## Additional files

### Supplementary files

• Supplementary file 1. fMRI cluster-based statistics.

• Transparent reporting form

### Data availability

The data has been made available on OpenNeuro (https://openneuro.org/datasets/ds002000/versions/1.0.0). Additionally, the data used to create the figures can be found on the Github repository with the associated scripts. (https://github.com/benjaminGriffiths/reinstatement_fidelity; copy archived at https://github.com/elifesciences-publications/reinstatement_fidelity).

The following dataset was generated:

| Author(s) | Year | Dataset title | Dataset URL | Database and Identifier |
|---|---|---|---|---|
| Griffiths BJ, Mayhew SD, Mullinger KJ, Jorge J, Charest I, Wimber M, Hanslmayr S | 2019 | Alpha/beta power decreases track the fidelity of stimulus-specific information | https://doi.org/10.18112/openneuro.ds002000.v1.0.0 | OpenNeuro, 10.18112/openneuro.ds002000.v1.0.0 |

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
