## [Decision Letter]

**Acceptance summary:**

This study addresses a timely and relevant question, namely, what is the functional role of alpha- and beta-band oscillations? Here, a novel approach is used to understand whether these oscillations modulate the processing of sensory information in the brain, as suggested (but not empirically tested) by a number of previous studies on the relationship between alpha/beta activity and neural and perceptual excitability. The authors use combined EEG-fMRI to independently estimate two neural measures: the alpha/beta power decrease and fMRI representational similarity analysis as a proxy of stimulus-specific information. They find that alpha/beta power decreases correlate with stimulus-specific information, both in the visual and auditory domains. The authors conclude that alpha/beta power decreases are a neural signature of the fidelity of stimulus-specific information.

**Decision letter after peer review:**

Thank you for submitting your article "Alpha/beta power decreases track the fidelity of stimulus-specific information" for consideration by *eLife*. Your article has been reviewed by three peer reviewers, and the evaluation has been overseen by Saskia Haegens as the Reviewing Editor and Laura Colgin as the Senior Editor. The following individuals involved in review of your submission have agreed to reveal their identity: Malte Woestmann (Reviewer #1); Nicholas Edward Myers (Reviewer #3).

The reviewers have discussed the reviews with one another and the Reviewing Editor has drafted this decision to help you prepare a revised submission.

Summary:

The authors analyzed data of a final sample of N=21 human participants in a combined fMRI-EEG study using an associative memory task. Stimulus-specific information, quantified via fMRI representational similarity analysis (RSA), was significantly correlated with concurrently recorded alpha/beta band power decreases. This effect was found for stimulus perception and episodic memory retrieval. The authors conclude that alpha/beta power decreases are a neural signature of the fidelity of stimulus-specific information.

The study addresses an interesting and timely research question and is well designed. We have a number of concerns that should be addressed before the manuscript can be considered for publication in *eLife*.

Essential revisions:

1) The authors should relate their results to the rich and recent literature regarding the functional role of alpha (and beta) power decreases for information processing. While some studies in the past might suggest that alpha power decreases reflect enhanced sensitivity (Hanslmayr et al., 2007; van Dijk et al., 2008; Busch et al., 2009), more recent studies have used sophisticated psychophysical modeling and neat experimental designs to demonstrate that alpha power decreases (prior to but also following stimulus onset) rather reflect enhanced neural baseline excitability, which affects subjective perception (Lange et al., 2013), confidence ratings (Samaha et al., 2017; Wostmann et al., 2019), response bias (Limbach et al., 2016; Benwell et al., 2017; Iemi et al., 2017), awareness (Benwell et al., 2017), and self-rated levels of attention (Whitmarsh et al., 2017) but not task accuracy.

- The authors interpret alpha/beta power decreases as a signature of the "fidelity of stimulus-specific information", which speaks to sensitivity rather than baseline excitability as the underlying neural mechanism. Unfortunately, results of the present study are difficult to integrate with the existing literature (listed above), since previous studies established relations between alpha/beta power decreases and different behavioral metrics whereas the present study relates alpha/beta power decreases mainly to results of the fMRI representational similarity analysis.

- Related to this point, the same correlation could be the result of a bimodal distribution of power (i.e. alpha/beta is either in a high or low power state, as suggested, e.g., by Freyer et al., 2009). If this distinction is important to the authors, it might be worth elaborating on this point in the Discussion.

- Although the authors made an attempt to relate alpha/beta power decreases to confidence ratings, a multivariate analysis (multiple regression or mediation analysis on single-trial data) would help to interpret the present results and integrate these with the existing literature. In order to investigate the exact relationship between RSA, alpha/beta power decreases and task performance in terms of confidence and accuracy, all of these variables would need to be combined in a statistical analysis. Results of such an analysis could also potentially enhance the impact of the Discussion section, which is somewhat vague at present.

2) The baseline window stretched up to the onset of the stimulus (-1000 to 0 ms). Given the relatively large number of cycles in the estimate, this baseline estimate could include stimulus-evoked power. It would be helpful to rule out this potential confound by moving the baseline window back so that it does not take into account post-stimulus signal. Otherwise there is an ambiguity in the results: It seems plausible that a higher stimulus-evoked response corresponds to better decoding. Since the baseline window might be more influenced by this response than the post-stimulus power estimate (500-1500 ms), this could lead to a negative correlation between power and decoding. This could alternatively be tested by testing for correlations between decoding and ERP amplitude.

3) Related to this point, a recent study (Iemi et al., 2019) showed that strong ERD is present in trials with strong pre-stimulus power whereas weak ERD is present in trials with weak prestimulus power. Accordingly, It is important in this study to understand what oscillatory estimate (pre or post) modulates stimulus-specific information. The authors could re-run their correlation analysis (alpha/beta power x stimulus-specific information) using only prestimulus power and only post-stimulus power. Additionally, if any result is found for the analysis of prestimulus power, it would be useful to visualize the temporal specificity of the effects (e.g. time-frequency plot).

4) The authors focus their analysis on alpha- and beta-band oscillations and use fixed bands for their analysis (in the 8-12 Hz and 13-30 Hz, respectively). It is important to highlight that a change in power in a specific band can be due to either an increase in genuine oscillations (periodic signal), and/or a change in the slope and/or offset of the aperiodic signal (Haller et al., 2018). Accordingly, we do not know whether i) these results are specific to the alpha/beta frequencies, or include other frequencies as well, and whether ii) these results are specific to the periodic and/or aperiodic signal (offset and slope). To address these questions, the authors could analyze the power spectrum in a broad frequency range (e.g. 2-50 Hz) and parameterize it into periodic and aperiodic signal, as suggested by Haller et al., 2018. Then they could assess their results for different aperiodic-adjusted frequency bands and for the slope and offset of the aperiodic signal.

5) How were BOLD responses to individual stimuli estimated? With a single-trial GLM? Was an HRF used, or a FIR basis set? Given the short interval between cue onset and choice screen onset in the retrieval phase, could some of the BOLD response be related to processing of the choice screen, and therefore partially confounded by the perceptual similarity between the chosen visual stimulus and the corresponding video?

6) Was BOLD amplitude correlated with alpha/beta power? It would be interesting to establish this before factoring BOLD amplitude out of the power/decoding correlation analysis. More details could also be given about this analysis - which voxels were selected for the BOLD amplitude estimate? Furthermore, it would be useful to visualize the full/partial correlations between alpha/beta power, fMRI BOLD signal, and fMRI stimulus-specific information in the manuscript.

---

## [Author Response]

Essential revisions:1) The authors should relate their results to the rich and recent literature regarding the functional role of alpha (and beta) power decreases for information processing. While some studies in the past might suggest that alpha power decreases reflect enhanced sensitivity (Hanslmayr et al., 2007; van Dijk et al., 2008; Busch et al., 2009), more recent studies have used sophisticated psychophysical modeling and neat experimental designs to demonstrate that alpha power decreases (prior to but also following stimulus onset) rather reflect enhanced neural baseline excitability, which affects subjective perception (Lange et al., 2013), confidence ratings (Samaha et al., 2017; Wostmann et al., 2019), response bias (Limbach et al., 2016; Benwell et al., 2017; Iemi et al., 2017), awareness (Benwell et al., 2017), and self-rated levels of attention (Whitmarsh et al., 2017) but not task accuracy.- The authors interpret alpha/beta power decreases as a signature of the "fidelity of stimulus-specific information", which speaks to sensitivity rather than baseline excitability as the underlying neural mechanism. Unfortunately, results of the present study are difficult to integrate with the existing literature (listed above), since previous studies established relations between alpha/beta power decreases and different behavioral metrics whereas the present study relates alpha/beta power decreases mainly to results of the fMRI representational similarity analysis.

We have now added a section in the Discussion to address this. We concur that our measure reflects sensitivity rather than baseline excitability, but do not feel that these results contradict these existing studies as these previous studies predominately consider pre-stimulus power and subjective measures of task performance, whereas we consider post-stimulus power and objective measures of perception. While the reviewers mention that the link between alpha power and criterion measures extends into the post-stimulus window, this is difficult to conclude based on the papers presented above. From our reckoning, only two present data from the post-stimulus window and they seemingly contradict one another (Benwell et al., 2017, shows a power decrease with increased accuracy [i.e. sensitivity; Figure 3B]; Wöstmann et al., 2019, shows a power decrease with increased confidence [i.e. criterion; Figure 2B; though notably not part of the significant cluster]). If one considers research into episodic memory, where d’ and ROC curves (measures of sensitivity) are often used to assess memory performance, one can see that post-stimulus alpha power indeed correlates with sensitivity (e.g. Fellner et al., 2019; Michelmann et al., 2016). This corroborates our findings, as well as the post-stimulus effects presented by Benwell et al., 2017. As for Wöstmann et al., 2019, it is worth noting that the effect they present seems more like an extension of the pre-stimulus effect rather than a stimulus-induced power decrease that we observed, Benwell et al., 2017, observes, and the memory studies above observe. As such, it would seem that while pre-stimulus power correlates with criterion, post-stimulus power more robustly correlates with sensitivity. We have devoted a paragraph in the Discussion to this.

“It is worth noting that we did not observe a correlation between stimulus-specific information and pre-stimulus alpha/beta power. […] As these previous studies have demonstrated consistently that pre-stimulus power does not correlate with measures of sensitivity, it is perhaps no surprise that we found no correlation between stimulus-specific information and pre-stimulus power.”

- Related to this point, the same correlation could be the result of a bimodal distribution of power (i.e. alpha/beta is either in a high or low power state, as suggested, e.g., by Freyer et al., 2009). If this distinction is important to the authors, it might be worth elaborating on this point in the Discussion.

We empirically addressed this issue on two fronts. First, we examined the distribution of alpha power across trials for each participant. Qualitatively speaking, it would seem that alpha power better approximates a normal rather than a bimodal distribution (see Figure 4—figure supplement 3C). Second, we re-ran our central analyses using a median split on our measure of power. If the original effect could be explained by a bimodal distribution of power, then splitting the power data into low and high power (approximating a bimodal distribution) should result in a similar correlation to the original effect. However, we found no evidence to suggest that a contrast of high and low power can explain similarity. We have referred to this in the main text and added full results as a supplementary figure (see Figure 4—figure supplement 3).

“It is worth noting that a bimodal distribution of power could explain these results (Freyer et al., 2009). […] Furthermore, visual inspection of the distribution of alpha/beta power within each participant suggests that alpha/beta power is normally distributed around the mean, rather than bi-modally distributed (see Figure 4—figure supplement 3).”

- Although the authors made an attempt to relate alpha/beta power decreases to confidence ratings, a multivariate analysis (multiple regression or mediation analysis on single-trial data) would help to interpret the present results and integrate these with the existing literature. In order to investigate the exact relationship between RSA, alpha/beta power decreases and task performance in terms of confidence and accuracy, all of these variables would need to be combined in a statistical analysis. Results of such an analysis could also potentially enhance the impact of the Discussion section, which is somewhat vague at present.

We have consolidated the correlations and partial correlations into a singular multiple regression (for perception and retrieval separately) investigating how power (both post-stimulus and pre-stimulus), confidence and BOLD influence measures of stimulus-specific information. The change in analysis does not drastically affect the results or their interpretability (see Figure 4). We have added panels in Figure 4 to depict how each predictor relates to the outcome variable of stimulus-specific information. The absence of a consistent correlation between stimulus-specific information and confidence (a subjective measure shown to correlate with baseline excitability) and BOLD amplitude (a neurophysiological measure of cortical excitability), supports the idea that stimulus-specific information is a measure of sensitivity and not one of baseline excitability. It is worth noting that this does push the link between post-stimulus alpha/beta power and stimulus-specific information into the ‘trending’ territory of significance (p = 0.054). As the retrieval effect remains strongly significant (p < 0.001) and the auditory perception analysis also yields a significant correlation (p = 0.006), we are not overly concerned by the trending nature of the visual perception analysis. We have elaborated on what may have caused this relatively small visual effect in the Discussion.

“Intriguingly, our analysis revealed that the correlation between alpha/beta power decreases and stimulus-specific information was weakest during visual perception. […] In short, the comparatively small effect in the visual perception task relative to the auditory perception and memory retrieval tasks may be explained by a limited variation in stimulus representation across trials.”

2) The baseline window stretched up to the onset of the stimulus (-1000 to 0 ms). Given the relatively large number of cycles in the estimate, this baseline estimate could include stimulus-evoked power. It would be helpful to rule out this potential confound by moving the baseline window back so that it does not take into account post-stimulus signal. Otherwise there is an ambiguity in the results: It seems plausible that a higher stimulus-evoked response corresponds to better decoding. Since the baseline window might be more influenced by this response than the post-stimulus power estimate (500-1500 ms), this could lead to a negative correlation between power and decoding. This could alternatively be tested by testing for correlations between decoding and ERP amplitude.

Yes, this is a valid concern. This has been addressed by shifting the baseline window to -1000 to -375ms. 375ms would be as far as we would expect smearing to extend given the use of 6-cycle wavelets to analyse 8Hz signal. This change has had no impact on the results.

3) Related to this point, a recent study (Iemi et al., 2019) showed that strong ERD is present in trials with strong pre-stimulus power whereas weak ERD is present in trials with weak prestimulus power. Accordingly, It is important in this study to understand what oscillatory estimate (pre or post) modulates stimulus-specific information. The authors could re-run their correlation analysis (alpha/beta power x stimulus-specific information) using only prestimulus power and only post-stimulus power. Additionally, if any result is found for the analysis of prestimulus power, it would be useful to visualize the temporal specificity of the effects (e.g. time-frequency plot).

We have split alpha/beta power into two regressors in the multiple regression, one for pre-stimulus power and one for post-stimulus power (being sure that temporal smearing from the ERP does not contaminate either metric). Regression analysis of this data suggests that it is post-stimulus rather than pre-stimulus power that predicts stimulus-specific information. This has been discussed throughout the last subheading of the Results. We have also added two figures to visualise the time-course and power-spectrum of the effect (see Figure 4C).

4) The authors focus their analysis on alpha- and beta-band oscillations and use fixed bands for their analysis (in the 8-12 Hz and 13-30 Hz, respectively). It is important to highlight that a change in power in a specific band can be due to either an increase in genuine oscillations (periodic signal), and/or a change in the slope and/or offset of the aperiodic signal (Haller et al., 2018). Accordingly, we do not know whether i) these results are specific to the alpha/beta frequencies, or include other frequencies as well, and whether ii) these results are specific to the periodic and/or aperiodic signal (offset and slope). To address these questions, the authors could analyze the power spectrum in a broad frequency range (e.g. 2-50 Hz) and parameterize it into periodic and aperiodic signal, as suggested by Haller et al., 2018. Then they could assess their results for different aperiodic-adjusted frequency bands and for the slope and offset of the aperiodic signal.

We completely agree that the original wavelet-based analysis confounds oscillatory activity with aperiodic activity. As such, we have added a supplementary analysis that uses the IRASA method to split the spectral activity into an oscillatory component, and an offset and slope parameter that characterises the aperiodic activity. We elected to use the IRASA method over the FOOOF method proposed by Haller et al. as the IRASA does not require the parameter tuning (and the many analytical paths that such tuning produces) that the FOOOF method does [see https://fooof-tools.github.io/fooof/auto_tutorials/plot_07-TroubleShooting.html#sphx-glr-auto-tutorials-plot-07-troubleshooting-py; accessed on October 2^nd^ 2019].

For the EEG event-related desynchronisation analysis, it seems that all three of these parameters can explain changes in wavelet-measured event-related reduction in power; see Figure 4—figure supplement 4). Specifically, we observed a decrease in oscillatory power and aperiodic slope, and an increase in aperiodic offset for the post-stimulus window relative to the pre-stimulus window (matching earlier findings; He et al. Neuron, 2010; Miller et al., 2009). In contrast, only a decrease in oscillatory power appeared to relate to successful memory retrieval (i.e. a decrease for hits relative to misses).

“It is also worth considering that functionally relevant changes in alpha/beta power needn’t necessarily reflect a change in oscillatory power, but rather a change in the fractal signal (Haller et al., 2018; Miller et al., 2009 (often referred to as the 1/f curve). […] As our central analysis was event-related and conducted on the single trial level, any approach we take would be curtailed by the biologically infeasibility of getting a robust estimate of the 1/f curve in such a brief time window.”

Notably, we could not identify any link between stimulus-specific information and periodic/aperiodic power. We think this is due to the fact that 1/f activity cannot be properly estimated for single-trial, event-related designs (see Figure 4—figure supplement 5). The 1/f curve is a product of numerous seconds of activity (Miller et al., 2009) – a period much longer than our 1 second window of analysis (itself based on previous studies; e.g. Michelmann et al., 2016). Therefore, a proper estimate of the 1/f cannot be obtained on a single trial, and instead must be averaged over a large number of trials (as done for the EEG analyses). Without a proper 1/f estimate, oscillatory power cannot be estimated either as oscillatory power is defined as the difference between the 1/f estimate and the raw power spectrum.As such, we cannot conclude whether the observed effect is due to changes in periodic or aperiodic signal (without a reliable measure, it would be unwise to interpret either a null or a positive result). However, critical appraisal of the EEG IRASA analyses suggests changes in oscillatory power are a better predictor for information representation – oscillatory power decreases transcend both task and sensory domain. In contrast, changes in aperiodic power only accompany stimulus onset and not memory retrieval – hence, they cannot explain information representation in memory tasks. It is also worth noting that the observed correlation between wavelet-based alpha/beta power and stimulus-specific information is frequency specific (Figure 4C) which is inconsistent with the idea that the observed link between alpha/beta power and stimulus-specific information is driven by changes in the 1/f slope.

5) How were BOLD responses to individual stimuli estimated? With a single-trial GLM? Was an HRF used, or a FIR basis set? Given the short interval between cue onset and choice screen onset in the retrieval phase, could some of the BOLD response be related to processing of the choice screen, and therefore partially confounded by the perceptual similarity between the chosen visual stimulus and the corresponding video?

In section 3, the BOLD responses for individual stimuli were estimated in GLM where each trial was entered as a separate regressor. As FIR basis set has the potential to fit to noise in such cases, a canonical HRF was used. We have clarified this in the Materials and methods.

“These trial-level BOLD responses were estimated using a GLM where each trial was considered as a separate regressor (using a stick function convolved with a canonical HRF as in the earlier analyses)”.

We agree that there is the possibility that the encoding-retrieval analysis is confounded by the stimulus selection window. To address this, we re-ran our encoding-retrieval similarity analysis during moment of selection (I.e. where confound was). If the reported encoding-retrieval similarity effect was driven by the stimulus selection screen, we would expect to see a similar spatial pattern (that is, bilateral fusiform activity) during this time-window. While two significant clusters survived thresholding (left lateral posterior temporal lobe: p_FWE_ = 0.002, k = 70, MNI [-54,-55,-6], d=0.12; left insula: p_FWE_ = 0.013, k = 49, MNI [-36,-4,-2], d=0.27), they did not overlap with the bilateral fusiform clusters observed in the original analysis; see Figure 2—figure supplement 2). Neither of these clusters survived the more stringent clustering threshold. These results would suggest that the BOLD response to processing/selecting the chosen stimulus does not explain the observed encoding-retrieval similarity effect observed bilaterally in the fusiform gyrus.

“Notably, this effect was not driven by the presentation of video stills that followed the presentation of the retrieval cue (see Figure 2—figure supplement 2).”

6) Was BOLD amplitude correlated with alpha/beta power? It would be interesting to establish this before factoring BOLD amplitude out of the power/decoding correlation analysis. More details could also be given about this analysis - which voxels were selected for the BOLD amplitude estimate? Furthermore, it would be useful to visualize the full/partial correlations between alpha/beta power, fMRI BOLD signal, and fMRI stimulus-specific information in the manuscript.

Yes, in Figure 4—figure supplement 1 we correlate the BOLD amplitude at every voxel with alpha/beta power (averaged across all electrodes) and identify a negative correlation between the two in the occipital lobe and cingulate cortex. Given the link between BOLD amplitude and alpha/beta power, we feel it is not only justified, but important to factor out the potentially-mediating factor of BOLD amplitude on the alpha/beta power * stimulus-specific information relation. We have further clarified how BOLD amplitude was calculated for the central multiple regression analysis. We have included a figure to depict the influence of each regressor (EEG metrics, BOLD amplitude, confidence) on stimulus-specific information (see Figure 4).

“BOLD activation was estimated for each trial, and was restricted to the same region of interest used to calculate stimulus-specific information and alpha/beta power in this analysis. […] The BOLD activation was then averaged across voxels within the ROI to provide a singular measure of BOLD activity for each trial.”